# Shift from Carbon Flow through the Microbial Loop to the Viral Shunt in Coastal Antarctic Waters during Austral Summer

**DOI:** 10.3390/microorganisms9020460

**Published:** 2021-02-23

**Authors:** Claire Evans, Joost Brandsma, Michael P. Meredith, David N. Thomas, Hugh J. Venables, David W. Pond, Corina P. D. Brussaard

**Affiliations:** 1Royal Netherlands Institute for Sea Research, P.O. Box 59, Den Burg, 1790 AB Texel, The Netherlands; JBrandsma@aceso-sepsis.org (J.B.); Corina.Brussaard@nioz.nl (C.P.D.B.); 2Ocean BioGeosciences, National Oceanography Centre, Southampton, European Way, Southampton SO14 3ZH, UK; 3Austere Environments Consortium for Enhanced Sepsis Outcomes, Henry M. Jackson Foundation for the Advancement of Military Medicine, Bethesda, MD 20814, USA; 4British Antarctic Survey, Natural Environmental Research Council, High Cross, Madingley Road, Cambridge CB3 0ET, UK; mmm@bas.ac.uk (M.P.M.); hjv@bas.ac.uk (H.J.V.); david.pond@stir.ac.uk (D.W.P.); 5Ecosystems & Environment, Faculty of Biological and Environmental Sciences, University of Helsinki, 00014 Helsinki, Finland; david.thomas@helsinki.fi; 6Faculty of Natural Sciences, University of Stirling, Stirling FK9 4LA, UK

**Keywords:** prokaryotes, Antarctica, viruses, heterotrophic nanoflagellates, microbial loop, viral shunt, carbon, bacteriovory, viral lysis

## Abstract

The relative flow of carbon through the viral shunt and the microbial loop is a pivotal factor controlling the contribution of secondary production to the food web and to rates of nutrient remineralization and respiration. The current study examines the significance of these processes in the coastal waters of the Antarctic during the productive austral summer months. Throughout the study a general trend towards lower bacterioplankton and heterotrophic nanoflagellate (HNF) abundances was observed, whereas virioplankton concentration increased. A corresponding decline of HNF grazing rates and shift towards viral production, indicative of viral infection, was measured. Carbon flow mediated by HNF grazing decreased by more than half from 5.7 µg C L^−1^ day^−1^ on average in December and January to 2.4 µg C L^−1^ day^−1^ in February. Conversely, carbon flow through the viral shunt increased substantially over the study from on average 0.9 µg C L^−1^ day^−1^ in December to 7.6 µg C L^−1^ day^−1^ in February. This study shows that functioning of the coastal Antarctic microbial community varied considerably over the productive summer months. In early summer, the system favors transfer of matter and energy to higher trophic levels via the microbial loop, however towards the end of summer carbon flow is redirected towards the viral shunt, causing a switch towards more recycling and therefore increased respiration and regeneration.

## 1. Introduction

The microbial loop describes the consumption of dissolved organic matter (DOM) by bacteria, which are then grazed by protists providing a route for carbon transfer to higher trophic levels [1,2]. The grazing step is mediated primarily by heterotrophic nanoflagellates (HNFs) which are the key bacteriovores in aquatic environments [3]. The microbial loop can serve alongside primary production as a carbon source to the food web [4]. Occurring in parallel is the ‘viral shunt’ whereby microbial hosts are infected and subsequently lysed by viruses, which converts their cell biomass to progeny and DOM [5]. In this way carbon is directed back into the pool of potential bacterial substrates [6]. Bacterial consumption of the DOM generated by the action of bacteriophages represents a closed trophic loop, whereby matter is recycled at the base of the food chain [7].

The amount of carbon from the DOM pool made available to higher trophic levels by the microbial loop will depend principally on the biotic and abiotic factors driving the magnitude of the fluxes of matter assimilated by bacteria in the first instance and bacteria consumed by grazers in the second. As bacterial production is controlled primarily by bottom-up processes (e.g., carbon availability [8]), factors that dictate DOM concentrations such as phytoplankton excretion and cell lysis [9] will have key significance. Bacterial prey availability will in turn influence carbon flow to HNFs, in addition to their specific grazing rates and levels of HNF standing stock, which are tightly linked to the intensity of predation on the HNFs themselves [10].

As with the microbial loop, the net effect of the viral shunt on carbon flow will depend on a complex web of factors related to the overlying physiochemical conditions [11] and the ecology of viruses, grazers, and their hosts. Theoretically, viral infection may serve to decrease carbon flow to higher trophic levels by reducing bacterial population size through cell lysis. Key factors in the net influence of viruses on carbon flow involve the ecology of viruses, bacteria, and grazers—abundance, susceptibility, virus infection strategy, succession rate, and grazing feeding strategies [7,12,13,14].

As the amount of carbon that flows through the microbial loop and the viral shunt is intimately tied to biotic and abiotic conditions we thus hypothesize that this is likely to be highly variable in regions subject to high seasonal fluctuations such as the polar marine waters. The Antarctic coastal regions show pronounced shifts in the levels of stocks and productivity of phytoplankton, bacterioplankton, and virioplankton [15,16,17,18]. However, as yet no attempt has been made to comprehensively study the seasonal dynamics of the microbial and viral mediation of carbon flow in this environment. We selected the surface waters of the coastal Western Antarctic Peninsula (WAP) as our study site given its well characterized fluctuations in environmental and biological conditions. The study was conducted during the biologically productive summer months.

## 2. Materials and Methods

**Study site and sampling.** The primary sampling site was located in the center of Ryder Bay (67.570° S 68.225° W), which lies at the northern end of Marguerite Bay on the Western Antarctic Peninsula (WAP; Figure 1). Seawater was collected from a depth of 15 m using a Niskin bottle deployed by winch from a rigid inflatable boat. Sampling was conducted from December 2010 until the end of February 2011, approximately weekly for process rates and twice weekly for abundances.

**Phytoplankton, bacterioplankton, viral and heterotrophic nanoflagellate abundances.** Chlorophyll concentrations were obtained using a WetLabs in-line fluorometer (see [15] for full details). For enumeration of viruses and bacteria, 2 mL seawater samples were fixed with a final concentration of 0.5% glutaraldehyde (EM grade, Sigma Aldrich, St. Louis, MO, USA) and stored at −80 °C. Viral and bacterial abundances were determined according to the methods of Mojica and colleagues [19] and Marie and colleagues [20], respectively. Briefly, samples were defrosted immediately prior to analysis, diluted with TE-buffer (pH 8.2) and stained with SYBR-Green I (Molecular Probes, Invitrogen, Waltham, MA, USA) at a final concentration of 0.5 × 10^−4^ (viruses) and 1 × 10^−4^ (bacteria) of the commercial stock. Staining was completed in the dark at 80 °C for 10 min for viral samples, and at room temperature for 15 min for bacterial samples. Counts were performed on a Becton-Dickinson FACSCalibur flow cytometer (BD Biosciences, Billerica, MA, USA). Groups were determined in bivariate scatter plots of green fluorescence of stained nucleic acids versus side scatter. For enumeration of heterotrophic nanoflagellates, triplicate (HNFs) 20 mL seawater samples were fixed for 15 min with glutaraldehyde at a final concentration of 2% (EM grade, Sigma Aldrich, St. Louis, MO, USA) and then stained with 4′,6-diamidino-2-phenylindole DAPI (Sigma Aldrich, St. Louis, MO, USA) at a final concentration of 1 µg mL^−1^ for 30 min in the dark. Samples were then filtered onto black 25 mm-diameter 0.2 µm polycarbonate filter with a 0.45-µm backing filter before being transferred to a microscope slide. Slides were stored at −20 °C in the dark until counting on an epifluorescent Zeiss Axioplan microscope (Axioplan II Imaging, Zeiss, Jena, Germany).

**Dissolved organic carbon (DOC).** Triplicate samples for DOC were collected by passage through syringe filters (Whatman GD/X GMF, pore size 0.45 µm) that had been rinsed thoroughly with sample water, before final sample filtrate was collected. The filtrates were then stored at −20 °C until later analysis. DOC was determined, after acidification of the samples using high temperature combustion using an MQ1001 TOC analyzer as described in Qian and Mopper [21]. International certified reference material (41 to 44 µmol L^−1^, Hansell Marine Laboratories, Miami, FL, USA) was used daily and compiled results for this were 42 ± 6 µmol L^−1^ (*n* = 261).

**Bacterial production.** Bacterial production was determined from the leucine incorporation rates based on the method of Simon and Azam (1989). Triplicate 10 mL samples and one control, killed by the addition of 0.5 mL concentrated formaldehyde, were inoculated with 40 nM ^3^H-leucine and incubated in the dark at in situ temperature for 4 h. Samples were then fixed by the addition of 0.5 mL concentrated formaldehyde and filtered onto 0.22-µm pore size, cellulose nitrate filters (Millipore HA). The filters were washed twice by the addition of 5% chilled trichloroacetic acid (TCA) for 5 min and then transferred into scintillation vials and stored at minus 80 °C until analysis. Prior to analysis 1 mL of ethyl acetate was added to the vials to dissolve the filters. After 10 min, 8 mL of scintillation cocktail (Packard Filter Count, PerkinElmer, Waltham, MA, USA) was added, and the samples were analyzed after 6 h on a Tri-Carb 2910TR liquid scintillation counter (PerkinElmer, Waltham, MA, USA). Heterotrophic bacterioplankton carbon production was calculated from leucine incorporation rates using the conversion factor of 1.5 kg mol^−1^ leucine incorporated [22,23].

**Lytic virus infection.** Viral production (VP) was determined by the viral reduction approach of Winget and colleagues [24]. Briefly, at in situ temperature and in low light conditions, a 600 mL whole seawater sample was reduced to 100 mL by recirculation over a 0.22-µm pore size polyether sulfone membrane (PES) tangential flow filter (Vivaflow 50, Vivascience, VWR, Leicestershire, UK)) at a filtrate expulsion rate of 40 mL min^−1^. Then, 600 mL of virus-free water generated by 30 kDa ultrafiltration using a PES membrane (Vivaflow 200, Vivascience, VWR, Leicestershire, UK) of an aliquot of the same seawater was then added to the whole water sample, which was again reduced to a volume of 100 mL by tangential flow filtration, and this was repeated three times. On the final flushing, filtration was stopped when the volume was reduced to approximately 500 mL. The sample was then gently agitated to mix and aliquoted into three 50 mL polycarbonate tubes pre-rinsed with the excess volume. Subsamples of 1 mL volume were immediately removed from each tube for viral and bacterial abundance (t0 samples). The tubes were incubated in darkness at in situ temperature and subsampled every 3 h for a further 9 h. Rates of lytic VP were determined from the slope of a first-order regression of virus concentration over time [25]. To derive a VP rate for the in situ community the experimental VP rate was corrected for the loss of bacteria resulting from the experimental set up.

**Bacteriovory.** Grazing on bacterioplankton by HNFs was determined using the fluorescently-labelled bacteria (FLB) disappearance method of Sherr and colleagues [26]. FLB were gently mixed with approximately 20 mL of seawater before being transferred to a 1.2 L polycarbonate bottle which was gently filled with whole seawater by siphoning. Experimental bottles were set up in triplicate and for each experiment a control of 0.2 µm filtered seawater was also set up. The final concentration of FLB was approximately 10% of the ambient bacterioplankton concentrations. Subsamples of 15 mL volume were taken immediately from all treatments and the bottles carefully filled with seawater to ensure they contained no air bubbles. The bottles were then incubated for 24 h on a slowly rotating wheel (0.5 rpm) at in situ temperature and light levels before being sampled again. The subsamples were stained with DAPI at a final concentration of 1 µg mL^−1^ for 30 min in the dark and filtered onto 0.2-µm black polycarbonate filters before assessing the abundance of bacteria and FLB using a Zeiss Axiophot epifluorescence microscope at the start and end of the experiment. FLB counts were corrected for the loss of the initial subsamples. We did not see change of FLB abundance in HNF-free samples over the 24 h incubation period. Rates of bacteriovory were calculated according to Salat and Marrasé [27], based on the specific net growth rate (a) and the specific grazing rate (g). For each triplicate net bacterial production (BPN) was determined according to:

BP_N_ = BA_0_ × (e^at^ − 1)

in cells mL^−1^ day^−1^, here BA_0_ is the bacterial abundance at the start of the experiment and t is the time of the experiment (1 day). The grazing rate (G) was then calculated by:
 G = (g/a) × BP_N_
in cells mL^−1^ day^−1^.

**Calculation of carbon budget.** Carbon content of the biogenic groups examined was calculated according to the conversion factors of 12.4 fg C cell^−1^ reported by Fukuda and colleagues [28] for bacterioplankton, the conservative estimate of 0.055 fg C particle^−1^ reported by Jover and colleagues [29] for virioplankton, and for HNFs, the estimate of 220 fg C µm^3^ reported by Børsheim and Bratbak [30] and the mean cell biovolume of 10 µm^3^ cell^−1^ reported by Baltar and colleagues [31].

Viral-mediated mortality (VMM) in terms of bacterioplankton lysed was calculated by dividing the rate of viral production by a burst size of 41 viruses cell^−1^ (Brum and colleagues [32]) as estimated from Antarctic Peninsula bacterioplankton. Carbon passing through the viral shunt was estimated based on the VMM of bacterioplankton multiplied by the conversion factors of 12.4 fg C cell^−1^ reported by Fukuda and colleagues [28]. Carbon passing through the viral shunt specifically directed into the virioplankton pool was based on the burst size of 41 viruses cell^−1^ reported by Brum and colleagues [32]. Grazing-mediated mortality (GMM) was estimated by dividing the specific grazing rate by the net bacterioplankton growth rate and then multiplying by net bacterial production. Carbon flowing to the DOM pool during heterotrophic nanoflagellate grazing was based on the estimate of 39% of total biomass consumed by Hagström and colleagues [33]. Our budget assumed that all the bacterioplankton biomass lysed by viruses is converted into either progeny viruses or DOM and is thus all shunted into the DOC pool.

## 3. Results

### 3.1. Abundances of Phytoplankton, Bacteria, Viruses, and Heterotrophic Nanoflagellates, and Concentrations of Dissolved Organic Carbon

At the onset of summer, the phytoplankton, as determined by chlorophyll a concentration (Figure 2A), began to bloom and reached a maximum by late December (18 µg L^−1^). The chlorophyll a concentration declined over the following two weeks, followed by another bloom during the second half of January (17 µg L^−1^). The decline of this bloom occurred at the end of January, after which chlorophyll a concentration increased to a plateau at around 7.5 µg L^−1^ for the majority of the remainder of the study. Bacterial abundance initially increased from 0.8 to 1.4 × 10^9^ cells L^−1^ over the first week in December but was followed by a decline to around 0.7 to 0.8 × 10^9^ cells L^−1^ during the month of January (Figure 2B). At the end of January, the bacterial abundance had further decreased to about 0.4–0.6 × 10^9^ cells L^−1^, and then increased again to 0.8 × 10^9^ cells L^−1^ by mid-February. Viral abundance overall increased over the course of the season, from 0.3 to 1.8 viruses L^−1^ (Figure 2C). More specifically, viral abundance peaked during the second half of January, coinciding with a decline in bacterial abundance (Figure 2B) and chlorophyll a (Figure 2A). Heterotrophic nanoflagellates abundance declined throughout the study period (Figure 2D), most markedly in December with concentrations falling from 3.1 to 1.2 × 10^6^ cells L^−1^. Thereafter the decline was more gradual reaching 0.5 × 10^6^ cells L^−1^ by the second half of February. DOC concentrations fluctuated between 99 and 149 µmol L^−1^ throughout the study with a general trend of higher concentrations in December and lower concentrations in February (Figure 2E).

### 3.2. Bacterial Production, Viral Production, and Grazing Rates

Bacterial production, as determined by leucine incorporation, showed three peaks over the study period (with maxima between 12 and 16 nmol L*^−^*^1^ d*^−^*^1^ (Figure 3A). Rates of viral production generally increased over the study period from minima of 0.3 to 2.7 × 10^10^ viruses L*^−^*^1^ day*^−^*^1^, with the exception of a low rate observed on the 31 January (Figure 3B). This low viral production rate matched the decline in viral abundance (Figure 2C). The peak periods of viral production followed bacterial production and chlorophyll a concentration. HNF grazing rates were highest in the first half of December and January (around 0.6 d*^−^*^1^), with low-rate intervals following (Figure 3C). After a third smaller peak beginning of February (maximum 0.4 d*^−^*^1^), rates declined to 0.2 d*^−^*^1^. Viral-mediated mortality (VMM) ranged between 0.8 and 6.5 *×* 10^8^ bacterial cells L*^−^*^1^ d*^−^*^1^, while grazing-mediated mortality (GMM) was between 1.1 and 8.3 *×* 10^8^ bacteria L*^−^*^1^ d*^−^*^1^ (Table 1). Expressed as % bacterial standing stock per day, VMM increased from a mere 6% in December to as high as 161% at the beginning of February. GMM showed a more consistent loss of between 20 and 74% bacterial standing stock d*^−^*^1^, with the lowest % in February. When considered as a % of the total available bacteria (i.e., standing stock and bacteria production), VMM ranged between 3 and 49%, while GMM ranged between 10 to 30%.

### 3.3. Budget of Carbon Flow

We estimated the carbon flux through the viral shunt and microbial loop (Figure 4) based on the three phytoplankton bloom periods (December, January, and February). There were clear differences between each month. The biogenic carbon in the bacterioplankton declined over the course of the season, from an average of 11 to 8 and 7 µg C L^−1^, respectively, for the three months examined (Figure 4). However, average monthly bacterial production was similar throughout the summer (11–12 µg C L^−1^ d^−1^). VMM increased on average more than 8-fold (0.9, 3.5, and 7.6 µg C L^−1^ d^−1^ for December, January, and February, respectively), while GMM decreased from around 5.8 µg C L^−1^ d^−1^ in December and January to 2.4 µg C L^−1^ d^−1^ in February (Table 1). The biogenic carbon in the HNF pool declined from on average 4.2 µg C L^−1^ in December to 1.9 µg C L^−1^ in January and reached a minimum of 1.4 µg C L^−1^ in February. In contrast, biogenic C in the virus pool was lowest in December (0.4 µg C L^−1^) and increased to 0.7 and 0.9 µg C L^−1^ in January and February, respectively.

## 4. Discussion

Over the course of the study the flow of carbon was observed to shift from a scenario where a large proportion passed through the microbial loop and little viral lysis occurred, towards a viral-shunt-dominated system where grazer-mediated transfer to higher trophic levels decreased by more than half. The dynamic nature of the activities and interactions of viruses, bacteria, and their grazers is unsurprising given the extreme physiochemical shifts that determine biological conditions in coastal Antarctic waters. Previous characterization of the study site has shown that the fluctuating chlorophyll concentrations are driven by light and iron availability which in turn are primarily controlled by the influence of sea ice on water column stability [17,34,35]. The levels of phytoplankton biomass accumulation and rates of primary production indirectly regulate bacterioplankton by determining substrate availability [18,36]. Thus, the fluctuations in bacterial production we observed, despite remaining on average similar over the three months of the study, were probably in large part due to the highly dynamic community of primary producers and thus levels of substrate available. The strength of secondary production varies substantially in the waters around Antarctica [26]. At the study site it provided a significant biogeochemical flux with high rates comparative to previous measurements at the WAP, but was consistent with findings that southerly, coastal regions having higher activity [8,16]. The temporal variations of DOC could be taken to suggest very active bacterial turnover of the labile and semi-labile DOC pool. The overall decline of bacterial standing stock, despite bacterial production remaining on average consistent, was likely due to increasing mortality pressure (sum of bacteriovory and viral lysis) throughout the study and was indicative of higher bacterial turnover in the latter part of the summer.

The highest grazing rates were observed in the first half of the study which is consistent with previous measurements in Antarctic waters indicating December as the point of maximal grazing [10,37]. Rates of bacteriovory are positively associated with abundances of HNFs [38] which, as with in other locations, have been shown to be coupled to bacterial abundance in the coastal Southern Ocean [39]. These observations fit with our own as both HNF and bacterial abundances declined over the study period, and the reduction of HNFs was likely due to a combination of bottom-up as well as top-down control mediated by increasing numbers of carnivorous zooplankton [10] or potentially also viral infection [40,41]. As with our own study, an inability to balance bacterioplankton production with bacteriovory has been reported for Antarctic waters (e.g., [42]) and the high rates of viral production in these areas [43,44] suggest that bacteriophage are also important for this role. Indeed, recently we have used an 8-year time series to demonstrate the close coupling of virioplankton to bacterioplankton at the study site [18], and Brum and colleagues [32] confirmed the presence of virally infected bacterioplankton at the coastal Antarctic. During our study viral-lysis rates likely rose in response to the increased host density and metabolic activity at the start of the summer [16]. This was probably further enhanced by the decrease in bacterioplankton diversity know to occur at this time [45,46], and which will have promoted viral infection [12]. These factors will also provide the trigger for the switch in dominant viral life cycles, from lysogenic to lytic, that occurs from December to January in coastal Antarctic waters [32], and which constitutes another factor that would have promoted lytic viral infection in the latter part of the summer.

Shifts in the relative significance of viral- and grazer-mediated mortality of bacterioplankton over time have been demonstrated previously in the East China Sea, where grazing prevailed in summer and viral lysis in autumn [38], and in the oligotrophic Mediterranean, where grazing was found to dominant over an initial year of measurements followed by a year where both pathways worked in concert [47]. It has been suggested that, while viral infection and grazing occurs in parallel in microbial communities, (and even simultaneously on the same host [48], their rates are generally inversely related [49]. Indeed, this has been shown to be the case in the Arctic at sites throughout the northern Greenland Sea and the Arctic Ocean [50]. Furthermore, it has been suggested that potentially both antagonistic and synergistic interactions may occur between these two sources of mortality [49,51]. It is, therefore, possible that the shifts we saw from a grazing-dominated to a viral-lysis-dominated system in the coastal Antarctic was facilitated to some degree by these interactions. As the shift we measured was from grazing to viral lysis this suggests that if interactions between these mortality pathways were in play, then they were likely to be grazer-mediated factors that promote viral lysis and/or viral-mediated factors that suppress grazing. A primary example of the latter was most likely the reduction in bacterioplankton abundance driven by the increasing levels of viral infection throughout the summer that would have likely increased the energy required for grazers to locate prey and therefore suppressed predation rates. Furthermore, as viruses are host-specific they may be responsible for driving succession in populations by limiting competitive dominance and thereby making niches available to other species [52]. As grazers may also feed selectively [53,54] compositional change of microbial populations due to viral activity could suppress grazing rates by forcing grazers to have to adapt to different prey types or to feed on suboptimal prey. Direct effects may also have been implicated such as the development of grazing-resistant cells either in response to bacteriovory [14,55,56] or after being infected by viruses [51]. It is highly probable that no one factor alone shaped the net levels of production or loss, and that at all times a web of factors (some of them contradictory) were in play. These will be mediated by the diversity of bacteria, viruses, and grazers, and the variety of strategies they employ in order to survive and reproduce. 

The net effect of the dynamics of microbial components and their loss and production terms throughout the summer indicated a dynamic community where the flow of carbon shifted substantially throughout the summer. During December, the microbial loop was the dominant fate of the carbon assimilated by bacteria indicating that in the early summer bacterioplankton make their most significant contribution to the Antarctic food web. Thus, carbon assimilated by heterotrophic bacteria during the early ‘spring’ bloom was most likely to ultimately support the production of higher trophic levels. With the shift towards viral-mediated mortality during the latter stages of the summer, the system switched towards a much greater level of carbon recycling and a reduced supply to higher trophic levels. This would have been further compounded by the decrease in bacterial growth efficiency and the respiration of organic matter by the conversion of a portion of the DOM consumed by bacteria to CO_2_ caused by viral activity [57]. Hence, repeated passage of matter through the viral shunt, as more likely occurred in late summer, would lead ultimately to the loss of organic matter from the system.

## 5. Conclusions

The Southern Ocean is considered a carbon sink, making processes which influence carbon channeling through its bio-hydrosphere an important target for understanding the mechanisms behind this. Furthermore, whilst the rapid warming in the Antarctic Peninsula has recently encountered a hiatus, it is predicted to continue [58] promoting a shift towards a microbially-dominated system [16,59]. Thus, a better understanding of the microbial communities of the coastal Peninsula Antarctica will be key to determining the Southern Ocean’s current role in the Earth system, and to forecasting how this might change in the future. Typically, efforts to understand microbially-mediated carbon flow through the productive periods have focused on its relative significance to primary production and have generally reported this to increase with progression of the season [36,39,60,61]. However, our data indicate the changing fate of bacterial production over the spring to autumn season should also be considered. Shifts in the dominant bacterial mortality mechanism from grazing toward viral lysis will influence the fate of bacterially-assimilated carbon with a tendency towards a higher proportion being respired towards the latter portion of the season. Furthermore, our data also highlight the importance of temporal, in addition to spatial, measurements when attempting to determine the functioning of seasonally-driven ecosystems.

## Figures and Tables

**Figure 1 microorganisms-09-00460-f001:**
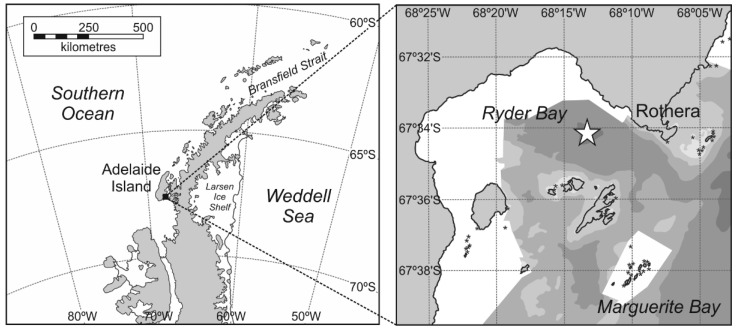
Map of the Western Antarctic Peninsula illustrating the location of the sampling site (white star), west of the British Antarctic Survey’s Rothera Research Station at 67.570° S 68.225° W.

**Figure 2 microorganisms-09-00460-f002:**
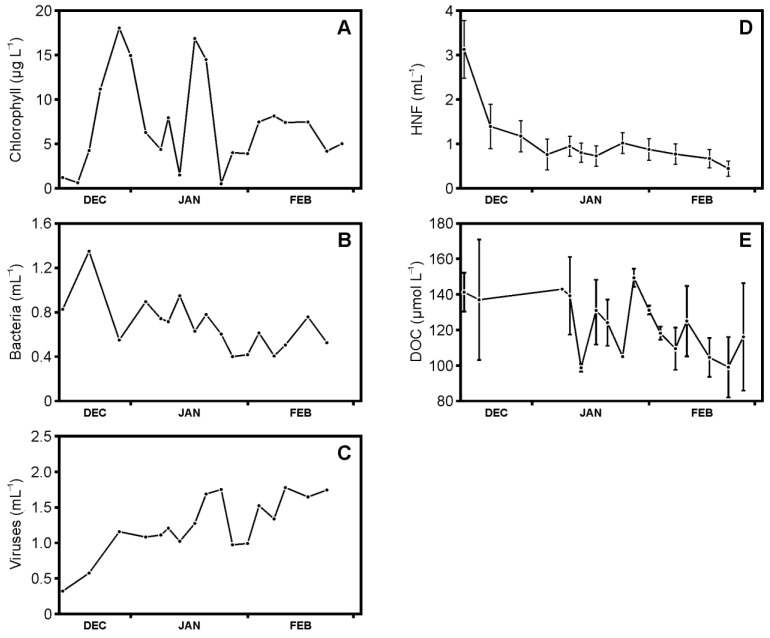
Plankton abundances and dissolved organic carbon (DOC) concentrations during the austral summer (December 2010 to February 2011) at the western Antarctic Peninsula sampling site at a water depth of 15 m: (**A**) chlorophyll concentrations are representative of phytoplankton levels, (**B**) abundances of bacterioplankton, (**C**) virioplankton, (**D**) heterotrophic nanoflagellates (HNFs), and (**E**) concentrations of dissolved organic carbon (DOC). HNF and DOC concentrations are averages of triplicate measurements with the error bar representing the standard deviation.

**Figure 3 microorganisms-09-00460-f003:**
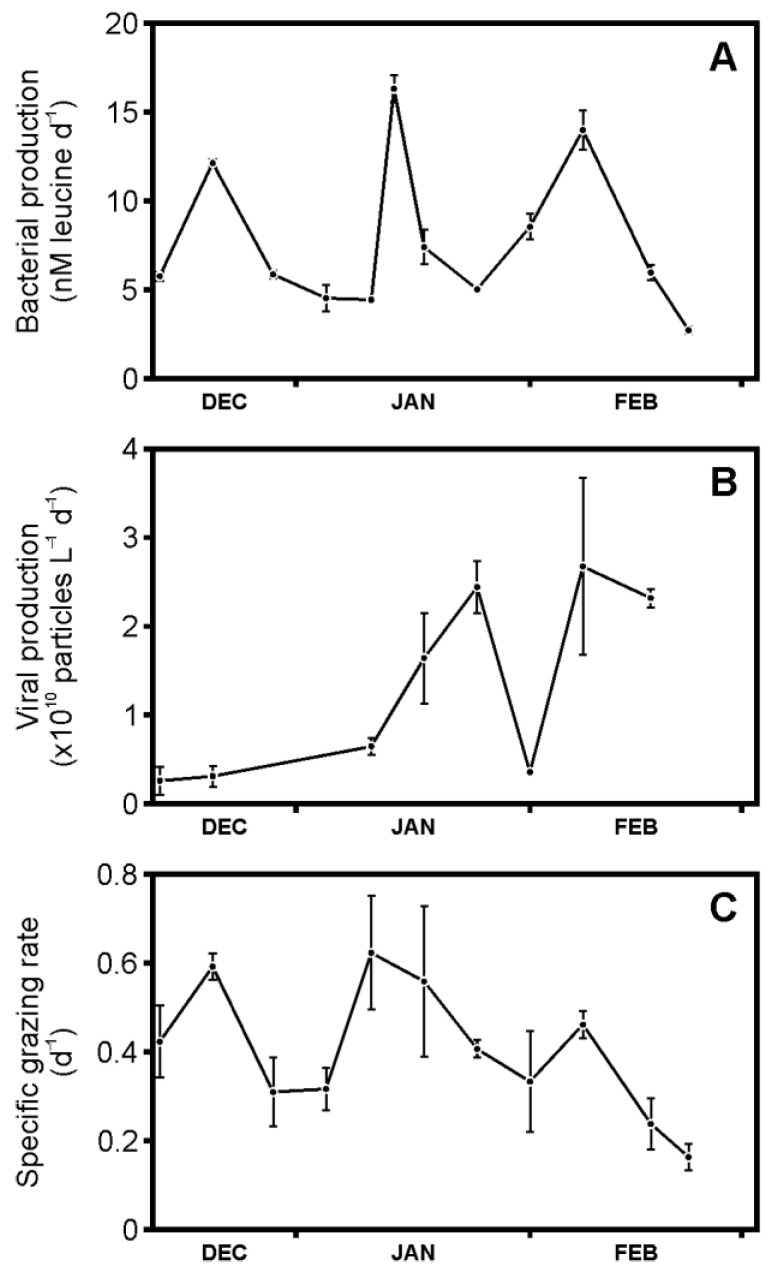
Rates of microbial processes mediating carbon flow during the austral summer (December 2010 to February 2011) at the western Antarctic Peninsula sampling site at a water depth of 15 m: (**A**) leucine incorporation, (**B**) viral production, and (**C**) specific heterotrophic nanoflagellate (HNF) grazing rates.

**Figure 4 microorganisms-09-00460-f004:**
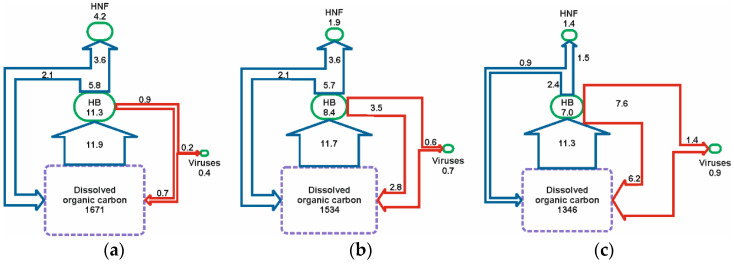
Diagrammatic representation of shifting carbon flow through the microbial loop (blue lines) and viral shunt (red lines) from: (**a**) December 2010, (**b**) to January 2011, and (**c**) to February 2011. Green boxes represent the carbon contained within the biogenic pools of heterotrophic bacterioplankton (HB), virioplankton, and heterotrophic nanoflagellates (HNFs) and values given have the units µg C L^−1^. Arrows represent the flux of carbon from one pool to another and values given have the units µg C L^−1^ day^−1^. Size of the boxes and thickness of the arrows is indicative of the magnitude of the carbon they represent with the exception of dissolved organic carbon (purple box), which, given its much greater magnitude, is not drawn to scale with the other budget components.

**Table 1 microorganisms-09-00460-t001:** Bacterial production and the loss of bacteria to viral and grazing-mediated mortality and their effects on carbon.

Date	Bacterial Production (×10^8^ Cells l^−1^ Day^−1^)	VMM (×10^8^ Cells l^−1^ Day^−1^) ^a,b^	GMM (×10^8^ Cells l^−1^ Day^−1^) ^a,b^	VMM (% of Bacterial Standing Stock d^−1^) ^a,b^	GMM (% of Bacterial Standing Stock d^−1^) ^a,b^	VMM (% of Total Available Bacteria d^−1^) ^a,b^	GMM (% of Total Available Bacteria d^−1^) ^a,b^	Bacterial Production Virally Lysed or Grazed (% d^−1^) ^a,b^	VMM (µg C l^−1^ d^−1^) ^c^	GMM (µg C l^−1^ d^−1^) ^c^
13 December 2010	7.0	0.6	3.9	7.7	46.9	4.2	25.4	64.6	0.8	4.8
20 December 2010	14.7	0.8	8.3	5.7	61.8	2.7	29.6	62.0	0.9	10.3
28 December 2010	7.1		1.7		31.5		13.7			2.1
04 January 2011	5.5		3.3		38.7		23.7			4.1
10 January 2011	5.4	1.6	5.8	20.1	74.2	11.9	44.1	138.1	2.0	7.2
14 January 2011	19.7									
24 January 2011	6.1	6.0	2.6	98.9	43.1	49.2	21.4	140.4	7.4	3.2
31 January 2011	10.4	0.9	1.4	21.0	34.3	6.0	9.8	22.2	1.1	1.8
07 February 2011	16.9	6.5	2.2	161.4	53.5	31.1	10.3	51.3	8.1	2.7
16 February 2011	7.2	5.7	1.9	74.7	25.5	38.2	13.1	105.1	7.0	2.4
21 February 2011	3.3		1.1		20.0		12.3			1.3

^a^ Abbreviations: VMM, virus-mediated mortality; GMM, grazing-mediated mortality; SS, standing stock; and BP, bacterial production. ^b^ VMM, viral production/burst size of 41 according to Brum and Colleagues [32]; GMM, (specific grazing rate/net growth rate) × net bacterial production. VMM of bacterial standing stock = (VMM/SS) × 100. GMM of bacterial SS = (GMM/SS) × 100. VMM of total available bacteria = [VMM/(SS + BP)] × 100. GMM of total available bacteria = [GMM/(SS + BP)] × 100. Bacterial production virally lysed or grazed = [(VMM + GMM)/BP] × 100. ^c^ Bacterial carbon conversion factor of 12.4 fg C cell^−1^ reported by Fukuda and colleagues [28].

## Data Availability

The data presented in this study are available on request from the corresponding author.

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
