# Peer review of "Shift from Carbon Flow through the Microbial Loop to the Viral Shunt in Coastal Antarctic Waters during Austral Summer"

_microorganisms, 2021, doi:10.3390/microorganisms9020460_

Round 1

Reviewer 1 Report

I was pleased to review "Shift from carbon flow through the microbial loop to the viral shunt in coastal Antarctic waters during austral summer" by Evans et al.

Overall, I found it to be a fantastic and informative read. I don’t believe I’ve read such a well-structured and concise examination of the microbial loop specifics prior to this manuscript.  The data, results and conclusions are all quite thorough, allowing for a complete picture from the authors’ investigations. After reading it several times over, I have very few comments and/or recommendations, and those are mostly minor, as the manuscript is quite sound.

Specifics:

Page 3, lines 87-88. The authors mention that two different references are used to determine abundances, one for viruses and one for bacteria. However, one of those references also includes methods for determining viral abundance. Why was a separate paper used for the viruses instead?

Page 3, line 90. The use of SYBR-Green will miss ssDNA and RNA viruses. I was hoping the authors would mention this issue to account for any inconsistencies in viral abundances.

Page 3, line 104. Just two things. There is an initial parenthesis before “(Whatman” but no closing parenthesis as there is only another open prior to (pore size and one closing after pore size. Also, the sentence itself seems a bit awkward, as if there might be a word missing. “had been rinsed with thoroughly with….” Was there supposed to be a word after “with”?

Page 3, line 115, A little clarification please, was this formaldehyde mentioned for the addition also using concentrated formaldehyde as indicated above in line 113? or different….

Table 1. Perhaps this issue will be solved during the formatting and printing process, but as the table stands now it is incredibly difficult to read. The headings are elongated awkwardly, the Dates are wrapping with hyphens, and trying to figure out what number goes with what requires a ruler. Could there be a way to adjust or even substitute info to make it clearer? Such as for Dates using numerical values instead of text (13/12/2010 vs. 13 December 2010).

Author Response

Reviewer 1

Overall, I found it to be a fantastic and informative read. I don’t believe I’ve read such a well-structured and concise examination of the microbial loop specifics prior to this manuscript.  The data, results and conclusions are all quite thorough, allowing for a complete picture from the authors’ investigations. After reading it several times over, I have very few comments and/or recommendations, and those are mostly minor, as the manuscript is quite sound.

We thank the Reviewer for these positive comments.

Specifics:

Page 3, lines 87-88. The authors mention that two different references are used to determine abundances, one for viruses and one for bacteria. However, one of those references also includes methods for determining viral abundance. Why was a separate paper used for the viruses instead?

The second paper, namely Mojica et al (2014), re-evaluated the methods used, and recommended to amend the original method of Marie et al (1999) by increasing the pH of the buffer used, given that this modification improved the sensitivity of the technique. Thus, we elected to use the amended protocol of Mojica et al, and so cited this reference.

Page 3, line 90. The use of SYBR-Green will miss ssDNA and RNA viruses. I was hoping the authors would mention this issue to account for any inconsistencies in viral abundances.

We kindly draw the Reviewer’s attention to the publication Brussaard et al. (2000) J Virol Methods which demonstrates that use of SYBR-Green I includes the ssDNA and RNA viruses.

Page 3, line 104. Just two things. There is an initial parenthesis before “(Whatman” but no closing parenthesis as there is only another open prior to (pore size and one closing after pore size. Also, the sentence itself seems a bit awkward, as if there might be a word missing. “had been rinsed with thoroughly with….” Was there supposed to be a word after “with”?

Thank you for drawing our attention to these errors, now corrected as follows: “Samples for DOC were passed through syringe filters (Whatman GD/X GMF, pore size 0.45 µm) that had been rinsed thoroughly with sample water, before final sample filtrate was collected.” 

Page 3, line 115, A little clarification please, was this formaldehyde mentioned for the addition also using concentrated formaldehyde as indicated above in line 113? or different….

Thank for drawing our attention to this omission. The missing information has now been included as follows: “Samples were then fixed by the addition of 0.5 ml concentrated formaldehyde…”

Table 1. Perhaps this issue will be solved during the formatting and printing process, but as the table stands now it is incredibly difficult to read. The headings are elongated awkwardly, the Dates are wrapping with hyphens, and trying to figure out what number goes with what requires a ruler. Could there be a way to adjust or even substitute info to make it clearer? Such as for Dates using numerical values instead of text (13/12/2010 vs. 13 December 2010).

We agree with the Reviewer regarding the presentation of the table and the subsequent implications for legibility and utility. We had hoped the table could run lengthwise on the page as opposed to width wise, however, we were unable to conduct this level of formatting on the template at this stage of submission. We have requested assistance from the editor to resolve this issue.

Reviewer 2 Report

The article from Evans et al. study the carbon flow in Antarctic coastal water during the summer, specifically the balance between the amount of Carbon transferred into higher trophic level by grazing of the bacterioplankton and the amount of carbon released back to the DOC pool by the viral shunt.

The article from Evans et al. is clear and efficient to make its point. This study brings interesting knowledge for a better understanding of the marine carbon flow in Antarctic waters. The study could have been stronger if the authors would have repeated the experiment at different location and compare the results, but the time series in itself is still relevant to show. Otherwise, the study would benefit from a clear hypothesis at the end of the introduction, and the authors could go a bit further in the discussion and discuss the impact of their finding on the overall system productivity for example.

Lines 63-69. Based on those finding in the literature, what would be the study hypothesis ? What could we expect in the WAP area given the highly variable nature of this environment ?

Line 76. Why choosing this particular depth ?

Lines 207-208 (Figure 2). Why is there only error bars in Figure 2E ? Is the error bar to small the be represented in the others graphs ?

Lines 256-262. Why are those different techniques used to calculate VMM and GMM ? What new or different information do they bring ? Those different number for the same process can be confusing.

Lines 324-326. Why does a decrease in bacterial diversity promote viral infection ? Please explain.

Line 355. Correct to ‘not’

Author Response

The article from Evans et al. study the carbon flow in Antarctic coastal water during the summer, specifically the balance between the amount of Carbon transferred into higher trophic level by grazing of the bacterioplankton and the amount of carbon released back to the DOC pool by the viral shunt.

The article from Evans et al. is clear and efficient to make its point. This study brings interesting knowledge for a better understanding of the marine carbon flow in Antarctic waters. The study could have been stronger if the authors would have repeated the experiment at different location and compare the results, but the time series in itself is still relevant to show. Otherwise, the study would benefit from a clear hypothesis at the end of the introduction, and the authors could go a bit further in the discussion and discuss the impact of their finding on the overall system productivity for example.

We thank Reviewer 2 for the positive and constructive review.

We agree wholeheartedly with the Reviewer regarding the virtue of the inclusion of additional sites, however attaining such data was not possible within the resources available for the study. Conducting a time series in Antarctic coastal waters had not been previously done to this detail and as such provides novel insights.  Performing research in Antarctica is a highly challenging undertaking. Travel between the bases is atypical and requires the charter of aircraft or ice strengthen-ships in an extremely remote part of the world. Moreover, comparing results to other locations during the same season would have entailed an additional extreme logistical and financial burden. Being able to make use of the logistics in place for the Rothera Long term timeseries by the British Antarctic Survey (BAS) allowed us to participate in their sampling trips benefiting from the massive investment of resources by BAS and their extensive operating experience in this extreme environment.

We have amended the introductory text as follows to more clearly sign post our hypothesis. As the amount of carbon that flows through the microbial loop and the viral shunt are intimately tied to biotic and abiotic conditions we thus hypothesize that they are likely to be highly variable in regions subject to high seasonal fluctuations such as the polar marine waters.” We appreciate that the Reviewer may prefer a more prescribed hypothesis, however a defined hypothesis is not obligatory and we feel to include such would not serve to improve the quality of the manuscript or be an effective mechanism to guide the reader. Specifically, we make the point that the strength of the microbial loop and viral shunt respectively are dependent on a web of complex factors, please see the preceding 2 paragraphs. “As bacterial production is controlled primarily by bottom-up processes (e.g., carbon availability [8]), factors that dictate DOM concentrations such as phytoplankton excretion and cell lysis [9] will have key significance. Bacterial prey availability will in turn influence carbon flow to HNF, in addition to their specific grazing rates and levels of HNF standing stock, which is tightly linked to the intensity of predation on the HNF themselves [10].” and “Key factors in the net influence of viruses on carbon flow involve the ecology of viruses, bacteria and grazers; abundance, susceptibility, virus infection strategy, succession rate and grazing feeding strategies [7, 12-14].” We do not feel it serves the flow of the text or the narrative to speculate how these complex factors might play out.

With regards to productivity we draw the reviewer’s attention to the final paragraph of the discussion, as follows and specifically the underlined text where we talk about the implications for productivity. “During December the microbial loop was the dominant fate of the carbon assimilated by bacteria indicating that in the early summer bacterioplankton make their most significant contribution to the Antarctic food web. Thus, carbon assimilated by heterotrophic bacteria during the early ‘spring’ bloom was most likely to ultimately support the production of higher trophic levels. With the shift towards viral mediated mortality during the latter stages of the summer, the system switched towards a much greater level of carbon recycling and a reduced supply to higher trophic levels. This would have been further compounded by the decrease in bacterial growth efficiency and the respiration of organic matter by the conversion of a portion of the DOM consumed by bacteria to CO2 caused by viral activity [57]. Hence, repeated passage of matter through the viral shunt, as more likely occurred in late summer, would lead ultimately to the loss of organic matter from the system.” Extrapolation beyond what is already given would necessitate significant conjecture, beyond that which can be supported by our data, and thus we feel this is beyond the scope of the manuscript.

Lines 63-69. Based on those finding in the literature, what would be the study hypothesis ? What could we expect in the WAP area given the highly variable nature of this environment ?

We kindly refer the Reviewer to our response above.

Line 76. Why choosing this particular depth ?

A depth of 15 m was used as is the standard depth employed by the long-term monitoring series at the British Antarctic Survey’s, Rothera site, at which we sampled. This allowed access to critical contextual supporting data (i.e. DOC, Chlo) beyond those collected in the central study (microbial state and rates) and facilitated our data’s contribution to improved understanding of a well characterised site, of long running and ongoing significance. It is also representative of the productive upper water column at which point greatest fluctuations in primary productivity occur, and thus the depth was of high suitability for our study.

Lines 207-208 (Figure 2). Why is there only error bars in Figure 2E ? Is the error bar to small the be represented in the others graphs ?

Chlorophyll concentrations (A) were determined from a sensor deployed in situ thus yielding only one measurement. Bacteria and viruses (B and C) are also determined from single samples given that the techniques employed have a high accuracy with the standard error between samples typically below 5%. For heterotrophic nanoflagellates concentrations (D) and DOC concentrations (E) triplicate samples were taken for each cast thus generating an Ave and SD. We have now included these details in the legend and made additions to the methods section as follows to ensure this information is clear to the reader. Line 95: For enumeration of heterotrophic nanoflagellates triplicate (HNF) 20 mL seawater samples were fixed for 15 min with glutaraldehyde at a final concentration of 2% (EM grade, Sigma Aldrich, USA) and then stained with DAPI (Sigma Aldrich, USA) at a final concentration of 1 µg mL-1 for 30 min in the dark.” Line 103: “Triplicate samples for DOC were collected by passage through syringe filters (Whatman GD/X GMF, pore size 0.45 µm) that had been rinsed thoroughly with sample water, before final sample filtrate was collected. The filtrates were then stored at –20°C until later analysis.” Included in the legend of Figure 2: “HNF and DOC concentrations are averages of triplicate measurements with the error bar representing the standard deviation.” We thank the Reviewer for drawing our attention to this point as we realise we had omitted the error bars from the DOC data now included on Figure 2. 

Lines 256-262. Why are those different techniques used to calculate VMM and GMM ? What new or different information do they bring ? Those different number for the same process can be confusing.

The manuscript’s remit is to examine the relative significance of the microbial loop and viral shunt. Table 1 serves to accomplish this by leading the reader through the relative significance of the key terms of these  different processes to the microbial food web. Specifically, the first three columns allow the relative strength of the production (bacterial production) and loss (VMM, GMM) processes to be understood by presenting these 3 rate terms side by side all expressed in the same units, namely cell per litre per day. To then examine the relative significance of the VMM and GMM processes in terms of the magnitude of the impact they have on the bacterial community it is necessary to place them into the context of how many bacteria there are (i.e. the standing tock). Specifically, this allows the reader to ascertain the relative magnitude of the two mortality terms overall. Put in simply do they kill a lot or a little of the bacteria. However, standing stock does not account for the fact that bacteria are continually growing and that the rate of growth further impacts the significance of these mortality terms. Put simply, if bacteria are growing very rapidly (irrespective of how many there are) then the relative significance of mortality rates will decrease. Or conversely even if there are lots of bacteria and mortality rates are low, death still maybe consider more significant if growth of the bacteria is low.  Thus, to express a relative measure of the overall mortality significance we also express VMM and GMM of total available bacteria (i.e. bacterial standing stock plus production) lost. The eighth column of the table, namely Bacterial production lysed or grazed allows the reader to determine if the bacterial abundance is in net growth or net decline. Finally, by converting mortality into carbon we present mortality rates in a common metric typically used as a currency to determine the flow of matter through food webs or through the Earth system, for example in computational models. This increases their utility for those readers who wish to integrate them into models or compare them with other terms such as primary production. It is typical to use tables of this sort to integrate different data stream of this nature.

Lines 324-326. Why does a decrease in bacterial diversity promote viral infection ? Please explain.

Viruses are host specific and dependent on passive encounters (mediated by Brownian motion) to infect their hosts. Therefore, the more diverse a potential host population the less likely viruses are to encounter a susceptible host. This is a well-established principal in viral ecology and we cite the appropriate reference specifically #12 now at page 327 of the revised manuscript. Thingstad, T.F.; Lignell, R. Theoretical models for the control of bacterial growth rate, abundance, diversity and carbon demand. Aquat. Microb. Ecol. 1997, 13, 19-27. doi: 10.3354/ame013019.

Line 355. Correct to ‘not’

We believe the reviewer must be referring to the sentence “As grazers may also feed selectively [53, 54] compositional change of microbial populations due to viral activity could suppress grazing rates by forcing grazers to have to adapt to different prey types or to feed on suboptimal prey.” which commenced on line 354 and terminates on line 356 of our original submitted manuscript. However, the sentence is correct and as intended.